# Policy Optimization with Second-Order Advantage Information

**Jiajin Li**[*], **Baoxiang Wang**[*]
Department of Computer Science and Engineering
The Chinese University of Hong Kong
{jjli,bxwang}@cse.cuhk.edu.hk

## Abstract

Policy optimization on high-dimensional action spaces exhibits its difficulty caused by the high variance of the policy gradient estimators. We present the action subspace dependent gradient (ASDG) estimator which incorporates the Rao-Blackwell theorem (RB) and Control Variates (CV) into a unified framework to reduce the variance. To invoke RB, the algorithm learns the underlying factorization structure among the action space based on the second-order gradient of the advantage function with respect to the action. Empirical studies demonstrate the performance improvement on high-dimensional synthetic settings and OpenAI Gym's MuJoCo continuous control tasks.

## 1 Introduction

Policy gradient methods Williams (1992) have been widely applied to various challenging problems including robotics Levine et al. (2016) and continuous control tasks Schulman et al. (2017); Lillicrap et al. (2015). The variance of the policy gradient estimator has been continuously improved by the Control Variates (CV) Oates et al. (2017) theory, that is, by subtracting a baseline function from the estimator. Examples are REINFORCE, Advantage actor-critic (A2C) Mnih et al. (2016), and action-dependent baselines Hao et al. (2018); Grathwohl et al. (2018). However, when dealing with high-dimensional action spaces, CV has limited effects regarding the sample efficiency. Rao-Blackwell theorem (RB) Casella & Robert (1996), though not heavily adopted in policy gradient, is commonly used with CV to address high-dimensional spaces Ranganath et al. (2014).

Motivated by the success of RB in high-dimensional spaces, we incorporate both RB and CV into a unified framework. We present action the subspace dependent gradient (ASDG) estimator. ASDG first splits the action domain and replace the expectation (i.e., policy gradient) with its conditional expectation over action subspaces (RB step). A baseline function associated with each of the corresponding action subspaces is used to further reduce the variance (CV step). While ASDG is benefited from both RB and CV's ability to reduce variance, we show that ASDG is unbiased under relatively weak model assumptions.

The major difficulty to invoke RB is to find a satisfying action domain partition. Novel trials such as Wu et al. (2018) utilize RB under the mean-field assumption which assumes that the policy distribution is fully factorized with respect to the action. Whilst it dramatically reduces the estimation variance, such a strong assumption restricts the policy distribution flexibility and thus conducts its optimization in a restricted domain. In our works, we show that Hessian of the advantage with respect to the action is theoretically connected with the action space structure. We exploit such second-order information with the minimum-cut algorithm to learn the underlying factorization structure in the action space. Instead of vanilla multi-layer perceptron, we utilize the wide & deep architecture Guo et al. (2017); Cheng et al. (2016) to capture such information explicitly. With the second-order advantage information, ASDG finds the partition that approximates the underlying structure of the action space.

We evaluate our method on a variety of reinforcement learning tasks, including a high-dimensional synthetic environment and several OpenAI Gym's MuJoCo continuous control environments. We

---

[*]These authors contribute equally to this work.

build ASDG on top of proximal policy optimization (PPO), and demonstrate that ASDG consistently obtains the ideal balance. While acquiring the sample efficiency introduced by RB Wu et al. (2018), it keeps the accuracy of the feasible solution Hao et al. (2018). As a result of such a balance, ASDG outperforms previous studies in high-dimensional tasks.

## 2 BACKGROUND & METHODS

We present our action subspace dependent gradient (ASDG) estimator by applying RB on top of the generalized action dependent baseline (GADB) estimator Hao et al. (2018); Grathwohl et al. (2018). We start from revisiting GADB. Denote $\pi_\theta(a|s)$ as the policy function, $c(s, a)$ as the action-dependent baseline function, $A^\pi(s, a)$ as the advantage function and $f_\theta(s, \xi)$ as the reparametrized policy function of $\pi_\theta(a|s)$. We have

$$\nabla_\theta J(\theta)_{GADB} = \mathbb{E}_\pi[\nabla_\theta \log \pi_\theta(a|s)\{A^\pi(s, a) - c(s, a)\} + \nabla_\theta f_\theta(s, \xi)\nabla_a c(s, a)].$$

To derive ASDG, we make the following model assumptions.

**Assumption 1 (Advantage Quadratic Approximation)** *Assume that the advantage function can be locally second-order Taylor expanded with respect to $a$ at some point $a^*$, that is,*

$$A^\pi(s, a) \approx A^\pi(s, a^*) + \nabla_a A^\pi(s, a)|_{a=a^*}^T(a - a^*) + \frac{1}{2}(a - a^*)^T \nabla_{aa} A^\pi(s, a)|_{a=a^*}(a - a^*).$$

*The baseline function $c(s, a)$ is chosen from the same family.*

**Assumption 2 (Block Diagonal Approximation)** *Assume that the row-switching transform of Hessian $\nabla_{aa} A^\pi(s, a)|_{a=a^*}$ can be approximated by the block diagonal matrix $diag(M_1, ..., M_k)$, where $\sum_{k=1}^{K} dim(M_k) = m$.*

Based on the assumption (1) and (2), the advantage function $A^\pi(s, a)$ can be divided into $K$ independent parts $A^\pi(s, a) = \sum_{k=1}^{K} A_k^\pi(s, a_{(k)})$, where $a_{(k)}$ denotes the subset of action dimensions to the $k$-th block $M_k$ and $a_{(-k)}$ is the complement of set $a_{(k)}$. The baseline function $c(s, a)$ is chosen to be divided in the same way.

**Theorem 3 (ASDG Estimator)** *If the advantage function $A^\pi(s, a)$ and the baseline function $c(s, a)$ satisfy assumption (1) and (2), the ASDG estimator $\nabla_\theta J(\theta)_{ASDG}$ is*

$$\sum_{k=1}^{K} \mathbb{E}_{\pi(a_{(k)}|s)}[\nabla_\theta \log \pi_\theta(a_{(k)}|s)(A^\pi(s, a_{(k)}) - c(s, (a_{(k)}, \tilde{a}_{(-k)}))) - \nabla_\theta f_\theta^k(s, \xi)\nabla_{a_{(k)}} c_k(s, a_{(k)})],$$

*where $\nabla_\theta f_\theta(s, \xi)$ is divided into $K$ parts as $[\nabla_\theta f_\theta^1, ..., \nabla_\theta f_\theta^K]$ and $\tilde{a}_{(-k)}$ is treated as a constant.*

Our assumptions are relatively weak compared with previous studies on variance reduction for policy optimization. We relax the fully factorization policy distribution assumption in Wu et al. (2018) to some constraints on the advantage function $A^\pi(s, a)$ with respect to the action space instead. Our assumptions are much easier to be satisfied, especially when the structure exhibited in the action space are captured by the Hessian matrix of the advantage function. Note that similar to previous works Wu et al. (2018), we just use the assumptions to obtain the structured factorization action subspaces and our estimator does not introduce additional bias.

**Connection with other works** - When we assume that the Hessian matrix of the advantage function has no block diagonal structure under any row switching transformation (i.e., $K = 1$), ASDG in Theorem.3 is the one inducted in Hao et al. (2018) and Grathwohl et al. (2018). Otherwise, if we assume that Hessian is diagonal (i.e., $K = m$), the baseline function $c(s, a_{(k)}, \tilde{a}_{(-k)})$ equals to $\sum_{i \neq k} c_i(s, a_{(i)})$, which means that each action dimension is independent with its baseline function. Thus, the estimator in Wu et al. (2018) is obtained.

**Using Generalized Advantage Estimator** - In this work, we obtain the realization value $\hat{A}(s, a)$ of advantage via Generalized Advantage Estimator (GAE) Schulman et al. (2015) $\hat{A}(s_t, a_t) = E[r_t + \gamma V^w(s_{t+1}) - V^w(s_t)]$, which can further significantly reduce the variance and avoid the action gap

at the cost of a small bias. Nevertheless, we cannot obtain the second-order information $\nabla_{aa}A(s,a)$ with the advantage realization value in the GAE identity. To address that, apart from the neural network $V^w(s_t)$, we employ a separate advantage network $A^\mu(s,a)$.

**Computing the partition** $a_{(k)}$ - Computing the second-order information directly from the network $\nabla_{aa}A^\mu(s,a)$ involves many problems: The Hessian matrix is both non-explicit and noisy. Specifically, the minimization of the first order error $||\hat{A}(s,a) - A^\mu(s,a)||^2$ does not necessarily guarantee the accurate second order information Li & Turner (2017). To address the problems, we utilize the recently proposed wide & deep architecture Guo et al. (2017) to estimate the advantage value. The architecture split the advantage function into two parts, including the quadratic term and the DNN component $A^\mu(s,a) = \beta_1 * A_{wide} + \beta_2 * A_{DNN}$, where $\beta_1$ and $\beta_2$ are the importance weights. Subsequently, we use Factorization Machine (FM) as our wide component $A_{wide}(s,a) = w_0(s) + w_1(s)^T a + w_2(s) w_2(s)^T \odot aa^T$, where $w_0(s)$, $w_1(s)$, and $w_2(s) \in \mathbb{R}^{m \times n_{fm}}$ are the action coefficients, $A \odot B = \sum_i \sum_j A_{ij} B_{ij}$ and $n_{fm}$ is the latent space size. To increase the signal-to-noise ratio of the second-order information, we approximate Hessian using $w_2(s) w_2(s)^T$.

## 3 EXPERIMENTS

Our algorithm is built on top of PPO where the advantage realization value is estimated by GAE. We have a policy network for PPO and a value network used by GAE that has the same architecture as in Mnih et al. (2016); Schulman et al. (2017).Our other parameters are consistent with those in Schulman et al. (2017) except that we reduce the learning rate by ten times for more stable comparisons. We design a synthetic environment with a wide range of action space dimensions and explicit action subspace structure to test the performance of ASDG and compare that with previous studies. Code is available at `https://github.com/wangbx66/Action-Subspace-Dependent`.

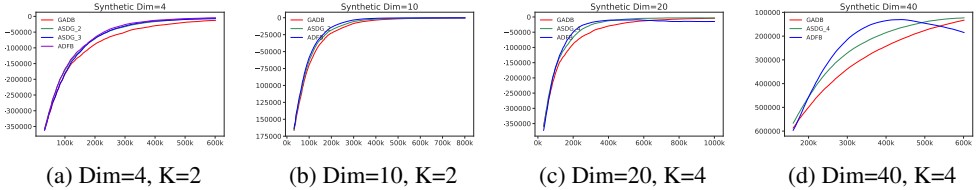

| (a) Dim=4, K=2 | (b) Dim=10, K=2 | (c) Dim=20, K=4 | (d) Dim=40, K=4 |

Figure 1: Learning curve for synthetic high-dimensional settings, varying from 4 to 40 dimensions.

We observe an ideal balance between accuracy and efficiency. On the one hand, ASDG trades marginal accuracy for efficiency when efficiency is the bottleneck of the training, as is in (a) and (b). On the other hand, ASDG trades marginal efficiency for accuracy when accuracy is relatively hard to achieve, as is in (c) and (d). ASDG's trade off results in the combination of both the merits of its extreme cases.

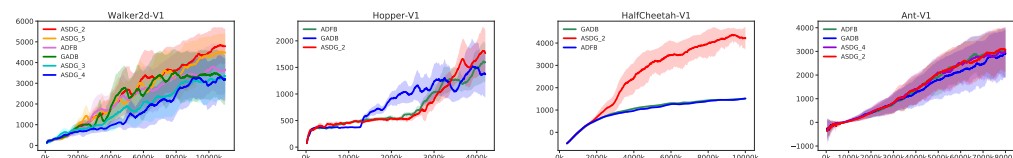

Figure 2: Comparison between two baselines Action dependent factorized baseline (ADFB) Wu et al. (2018) as well GADB and our ASDG estimator on various OpenAI Gym Mujoco continuous control tasks, including Walker2d-V1 (Dim=6), Hopper-V1 (Dim=3), HalfCheetah-V1 (Dim=6) and Ant-V1 (Dim=8).

We test ASDG on several environments with relatively high action dimensions, namely Walker2d, Hopper, HalfCheetah, and Ant, shown in Fig.2. In general, ASDG outperforms ADFB and GADB consistently but performs extraordinarily well when the action space satisfies structured mean field assumption such as in HalfCheetah. To discuss the choice of $K$, we test all the possible $K$ values in Walker. As discussed in the synthetic experiments, the optimal $K$ value is supposed to be between its extreme $K = 1$ and $K = m$ cases. Empirically, we find it effective to conduct a grid search. We consider the automatically approach to finding the optimal $K$ value an interesting future work.

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
