# OpenReview forum: "Policy Optimization with Second-Order Advantage Information"
_ICLR.cc/2018/Workshop — Accept_

### Official Review · AnonReviewer2 · 2018-03-09
**A good exploratory work on variance reduction in policy gradient, but lacks some empirical validations**

**Rating:** 6
**Confidence:** 5

**Review:**

The paper proposes using row-shifting approximate block diagonalization of advantage estimate to identity uncorrelated action subspaces to apply the methods in Wu et. al. (2018) over subspaces. It shows that it leads to some improvements in policy gradient performances.

The method proposed in the paper is closest to Wu et. al. (2018), except it identifies axis-aligned subspaces using quadratic advantage approximations. The method is a good addition to a list of control variates/baseline methods for policy gradient, and I recommend for acceptance. Q-Prop [Gu et. al., 2017] should also be added as a reference.

The author comment discusses Tucker et. al. (2018) results and suggests it’s mainly on control variates. This isn’t perfectly accurate, since the main discussion in their paper is adding extra single step action dependency in baseline (where control variate subsumes Wu et. al. 2018, except practical performance differences in function approximations) does not reduce variance significantly enough in some domains beyond learning a better baseline. Empirical estimations of the variance reductions are expected as follow-up.

Questions:
- Parameterization of c(s, a) and c_k(s,a_k) could be better clarified. Isn’t block diagonalization/row shifting different per state sample s? If so, it seems unclear how c is parameterized.

Pros:
- Clear description of the method

Cons:
- Performance improvements aren’t significant. In Figure 2, good halfcheetah results are enough, since they are notoriously high variance
- Empirical validation of variance reduction is required to complete the paper

---

### Official Review · AnonReviewer1 · 2018-03-11
**Interesting contribution, not sure about experimental evidence**

**Rating:** 6
**Confidence:** 3

**Review:**

Summary:
The goal of the paper is to reduce the variance of policy gradients methods when the action space is high dimensional.
To do so, they combine the idea of having action-dependent baselines (Generalized Action Dependent Baseline, Hao et al. 2018) with the idea of utilising independence between action dimensions (or groups thereof, i.e. factors, Wu et al. 2018) to have a separate baseline function for each factor.
They propose a method to learn an underlying factorisation structure, with the number of independent factors being a hyperparameter.

Novelty (6/10):
Most of the ideas in this paper have been previously published, in particular, both the RB and CV step.
They state that Wu et al. assumed a fully factorized policy, however in their paper, Appendix E, Wu et al. 2018 also discuss more general factorizations of the action.
What is new in this context is to automatically learn the factorization, given the number of independent factors. They do so by estimating the Hessian and applying the minimum cut algorithm.

Clarity (6/10):
The paper builds upon a great variety of previous work, which makes it difficult to understand it on its own without consulting some of the referenced papers as there is not enough space for a sufficiently detailed background section.
Overall, the paper explains the method sufficiently well, but could be structured better: For example, the usage of the minimum-cut algorithm to compute the partition based on the Hessian is only mentioned in the introduction but not outlined or even mentioned in the methods section.
I did not understand the sentence "ASDG's trade-off results in the combination of both the merits of its extreme cases" which was used to explain the superior performance in the experiment section on the synthetic task.

Significance (8/10):
Research around the question of how to reduce the variance of policy gradient algorithms is highly active at the moment. This paper fits nicely into this.

Quality (5/10):
- (+) Code is given
- (-) No proofs are given
- (-) There are inconsistencies between their reproduced results for ADFB (Wu et al.), which they use as baseline, and the results in the original paper: The results roughly agree for Ant, but are wastly better for HalfCheetah and Hopper in the original paper than they are reported here (~4000 vs ~1000) and (~3500 vs ~1500).
- Why is the performance of ADFB in the synthetic task decreasing over time (for experiments c) and d) )?

Pros:
- Highly relevant topic
- Interesting contribution on how to find a good factorisation of the policy that can be used for variance reduction

Cons:
- Combination of a lot of different ideas makes it difficult to distil the effect of their original contribution. For example, if one sees the learned factorization as their main original contribution, then I'm not sure why the CV step (Hao et al.) is necessary?
- Inconsistencies in baseline performance (see 'Quality' section).
- Lacking proofs and detailed description of their method (likely due to their combination of many different ideas)

---

### Official Review · AnonReviewer3 · 2018-03-12

**Rating:** 7
**Confidence:** 2

**Review:**

This paper essentially combines two ideas - the first is reparametrized action-conditional baselines, the second is to make block-diagonal assumption on the advantage function (instead of fully diagonal). While neither of the ideas appears fundamentally novel (and we recover as a special cases both ideas- ADFB and GADB), the combination is still worthwhile to investigate. Writing and notation in general could be improved (The 'computing the partition' section in particular is hard to parse). Experiments demonstrate moderate improvements over the baselines (ADFB/GADB).

Minor:
- 'Otherwise, if we assume that Hessian is diagonal... equals to..." : is this accurate? This results does not appear in Wu et al. afaik, and writing a baseline as a sum of baseline feels like a mistake (all are effectively conditional expectations of the same quantity, so the sum of some shouldn't equal another). Or did I misread the notation?

---

### Public Comment · ~Baoxiang_Wang1 · 2018-03-08
**Regarding the recent study "The Mirage of Action-Dependent Baselines in Reinforcement Learning"**

A recent paper is discussing the Mirage [1] of the "action-depend baselines". We'd like to note that the paper is discussing the ineffectiveness of action-dependent control variate (CV) [2,3]. Adding the action-dependent Rao-Blackwellization (RB) on top of CV [4 and our paper] is still a very promising improvement. Also, our algorithm and theory are straightforwardly extended to both action-dependent or independent (only state-dependent) CV. We also find in our experiment that we have similar performance using either action-dependent CV (baseline) or just state-dependent baseline (A2C).

[1] Tucker, George, et al. "The Mirage of Action-Dependent Baselines in Reinforcement Learning." arXiv preprint arXiv:1802.10031 (2018).
[2] Liu, Hao, et al. "Action-dependent Control Variates for Policy Optimization via Stein Identity."  ICLR 2018.
[3] Grathwohl, Will, et al. "Backpropagation through the Void: Optimizing control variates for black-box gradient estimation."  ICLR 2018.
[4] Wu, Cathy, et al. "Variance Reduction for Policy Gradient with Action-Dependent Factorized Baselines." ICLR 2018.

---

### Decision · Program_Chairs · 2018-03-20
**ICLR 2018 Workshop Acceptance Decision**

**Decision:**

Accept

**Comment:**

Congratulations, your paper was accepted to the ICLR workshop.